# Preliminary Evaluation of a Blast Sprayer Controlled by Pulse-Width-Modulated Nozzles

**DOI:** 10.3390/s22134924

**Published:** 2022-06-29

**Authors:** Enrique Ortí, Andrés Cuenca, Montano Pérez, Antonio Torregrosa, Coral Ortiz, Francisco Rovira-Más

**Affiliations:** Departamento de Ingeniería Rural y Agroalimentaria, Universitat Politècnica de València, 46022 Valencia, Spain; eorti@dmta.upv.es (E.O.); ancuecu1@upv.es (A.C.); montano@dmta.upv.es (M.P.); torregro@dmta.upv.es (A.T.); frovira@dmta.upv.es (F.R.-M.)

**Keywords:** precision spraying, blast sprayer, PWM nozzles

## Abstract

Precision spraying relies on the response of the spraying equipment to the features of the targeted canopy. PWM technology manages the flow rate using a set of electronically actuated solenoid valves to regulate flow rate at the nozzle level. Previous studies have found that PWM systems may deliver incorrect flow rates. The objective of the present study was to characterize the performance of a commercial blast sprayer modified with pulse-width-modulated nozzles under laboratory conditions, as a preliminary step before its further field validation. Four different duty cycles (25 percent, 50 percent, 75 percent and 100 percent) and four different pressures (400 kPa, 500 kPa, 600 kPa and 700 kPa) were combined to experimentally measure the flow rate of each nozzle. Results showed that the PWM nozzles mounted in the commercial blast sprayer, under static conditions, were capable of modulating flow rate according to the duty cycle. However, the reduction of flow rates for the tested duty cycles according to pressure was lower than the percentage expected. A good linear relation was found between the pressure registered by the control system feedback sensor and the pressure measured by a reference conventional manometer located after the pump. High-speed video recordings confirmed the accurate opening and closing of the nozzles according to the duty cycle; however, substantial pressure variations were found at nozzle level. Further research to establish the general suitability of PWM systems for regulating nozzle flow rates in blast sprayers without modifying the system pressure still remains to be addressed.

## 1. Introduction

It is generally assumed that the use of pesticides is necessary to ensure high crop yields providing enough food supply for people and high quality of food products [1]. However, it has become widely acknowledged that they can give rise to a range of side effects related to their toxicity, having a negative effect on humans and the environment [2,3]. As a result, various mitigation strategies have been proposed to reduce the negative effect of pesticides. In 2006, the authors in [4] reviewed 180 different publications dealing with mitigation strategies. However, they indicated that more research was required on possible interactions between different drift mitigation measures and the resulting drift reduction efficiency. Spray drift is defined as the movement of airborne spray droplets beyond the area of the application [5]. Orchard pesticide applications have been considered a prime source of spray drift and environmental contamination. Several reports have shown the significant effect of off-target drifts [6,7,8]. It has been reported that only 30–40 percent of the pesticide droplets deposit on the target, and therefore minimizing the emission and drift of pesticides for spray applications is vital [9]. In order to reduce the spray drift, mathematical and computational studies have been carried out to model the transportation of spray droplets [10].

Adapting the spraying equipment to the canopy characteristics has been proposed, to reduce drift [11,12]. Compared to conventional application techniques, variable spray technology, as a practical implementation of precision spraying, has become a promising approach to improve the use of pesticides and reduce their negative effect on humans and the environment [13]. Precision spraying relies on target information related to the tree canopy (size, shape, structure and density) gathered from various sensors such as cameras, ultrasound and lasers [14] and the response of the spraying equipment to this canopy information. Previous studies in [15] pointed out three steps to improve variable-rate spraying technology: (1) advanced real-time sensing technology to measure canopy volume; (2) accurate control algorithms for liquid flow rate and airflow regulation; and (3) design of variable-rate sprayers with liquid and air regulation. For citrus orchards, in particular, conventional air-assisted sprayers with hydraulic nozzles and a powerful air jet generated by a radial fan to carry the spray droplets are commonly used. Spraying operations with air-assisted equipment often produce significant off-target losses and high power consumption; therefore, optimized designs for airflow profiles could reduce off-target depositions [16,17].

The flow rate of spraying liquid at a given instant is determined by a general model that contains the desired spray volume, the calculation model and the decision model. Most existing spray volume models are based on the working area, the leaf wall height in relation to the tree height and planting row space, the leaf wall area according to the tree height and the tree row volume, assuming that the tree row is a cuboid canopy volume. Flow rate regulation can be achieved, for instance, by using PID-based pipeline adjustment and spray nozzle flow control with pulse width modulation of solenoid valves (PWM). PWM technology manages flow rates by varying the duty cycle [18] and using electronically actuated solenoid valves at nozzle level. It is a variable-rate technology with great potential to minimize the application errors caused by varying pressures during operation, as it manages the flow rate at nozzle level using independent solenoid valves. Compared to other ways of modifying flow rates during spraying tasks, PWM systems maintain the application pressure during the entire task, keeping droplet sizes within reasonable ranges. Nonetheless, several authors have reported significant variations in flow rate and droplet size with such systems. In [19], an experiment using 40 nozzles, each coupled with a PWM solenoid valve, showed that the operating pressure in the spray line dropped considerably as the duty cycle increased under no-pressure conditions, resulting in significantly lower flow rates compared to the pressure conditions.

In order to assess pressure and droplet size uniformity in PWM technology, the authors in [18] carried out field tests with a self-propelled hydraulic sprayer with 73 nozzles. In order to measure the real-time boom pressure during operation, 10 nozzles were selected across the sprayer boom. Their solenoid valves were removed, and the nozzle bodies capped to fit high frequency (under 1 ms response time) pressure transducers with an error under 0.25 percent at full scale. The overall system flow rate was measured using a turbine-type flow meter. The uniformity of the droplet size distribution was calculated by transforming the monitored pressure data to their equivalent droplet sizes, according to the manufacturer’s catalog. The results indicated that the PWM system was able to provide a uniform pressure, even under different field operating conditions. However, they recommended using water-sensitive cards to further evaluate the actual droplet size distribution. The same research team [20] studied three different PWM nozzle control systems and recorded nozzle pressure, boom pressure, flow rate and response time for different duty cycles. Significant differences were found between the three systems in pressure drop, stabilized pressure application time and flow rate. In addition, they found that PWM systems could deliver incorrect flow rates due to variations in the control application time cycle, as determined by three morphological parameters: peak time, stabilized application pressure time and fall time.

In order to assess precision variable-rate spraying in greenhouses, the authors in [21] carried out laboratory experiments using a high-speed laser scanning sensor, 12 PWM nozzles and a mobile spray boom controlled by a computer. The plants were simulated with four types of regularly shaped objects and two artificial plants. A high-speed camera was used to determine the delay times and nozzle activation. The accuracy of the spray control system with respect to spray delay time, nozzle activation and spray volume was evaluated according to object size, resulting in the reduction of spray volume, as the nozzles were precisely activated by the object presence. Regarding orchard applications, the authors in [22] conducted spraying tests in apple trees using a blast sprayer with PWM nozzle control. In particular, three different systems were compared: laser-controlled PWM, manually-controlled PWM and constant-rate spraying without PWM control. Spray coverage and deposition within canopies, as well as off-target losses, were measured using stainless steel screens and water-sensitive paper. The highest deposition amounts were registered for the system without PWM control, exceeding the optimal values and thus wasting part of the spray product. In relation to ground losses, laser-driven PWM produced the lowest mean ground losses, the smallest deviations and the shortest differences between rows, and was thus the most efficient spraying system of the three evaluated. The authors concluded that it was possible to improve application efficiency of air-blast sprayers by coupling the PWM valves to the nozzles during the treatment of young apple trees. Similarly, the authors in [23] studied, under laboratory conditions, the effects of duty cycles (from 10 percent to 100 percent at intervals of 10 percent) and standard flat fan nozzles (six) with and without nozzle operating pressure adjustments, measuring droplet size distributions and operating pressures. They found that the nozzles coupled with PWM solenoid valves could lead to a pressure increase across the duty cycle range, but this increase did not affect the diameter of the droplets. In another experiment, in [24], droplet size distributions of hollow-cone nozzles regulated by PWM valves at different working pressures (276, 414, 552, 689 and 827 kPa) were evaluated, with nozzle pressure, flow rate and droplet size distributions determined for different duty cycles. A laboratory spraying system was assembled to study five disc-core hollow-cone nozzles connected to a PWM solenoid valve, where two high frequency pressure transducers (less than 1 ms response time) were mounted upstream and downstream of the valve. The results indicated that smaller nozzles showed lower changes in pressure, flow rate, spray angle and volumetric diameter as duty cycles decreased and nozzle tip pressure increased. The upstream pressure at the PWM valve increased as the duty cycles decreased, with values of around 50 kPa. Additionally, greater downstream pressure reduction was found for higher pressures and larger nozzles when duty cycles decreased from 100 percent to 10 percent.

In this regard, the authors in [25] identified several concerns related to the effects of on-off solenoid valves and the alternating on-off action of adjacent nozzles on uneven spray coverage, specifically the effects of the PWM duty cycle and sprayer forward speed on the variability of spray coverage under standardized and actual field conditions for an espalier-trained vineyard. They used water-sensitive paper to register coverage variability along the sprayer track, the canopy height and the canopy depth. Before the field tests, laboratory tests were conducted to characterize the effect of the PWM duty cycle on the droplet size. A prototype vineyard sprayer with 12 configurations was tested (duty cycles: 30, 50, 70 and 100 percent; forward speeds: 4.0, 6.0 and 8.0 km/h), confirming that PWM spray systems may be an effective technology for variable-rate spraying in vineyards, although the pulsing effect of the PWM valves on the homogeneity of spray coverage was not studied under actual field conditions.

The objective of this paper is to characterize the performance of a commercial blast sprayer modified with pulse-width-modulated nozzles under laboratory conditions, testing the effects of duty cycles and pressure settings on flow rate and pressure at different circuit points. This experiment represents a preliminary step before the application of prescription maps featuring VRT actuation in vineyards and olive groves for sustainable spraying and efficient crop protection.

## 2. Materials and Methods

### 2.1. Materials

A commercial air-blast trailed sprayer for olive orchards was evaluated under static conditions in a laboratory. Off-the-shelf hollow disc-cone nozzle bodies (D10 disc and DC45 core, TeeJet Technologies, Glendale Heights, IL, USA) were used with solenoid shut-off valves (115880, TeeJet Technologies, Glendale Heights, IL, USA) for PWM control on each nozzle body, closing and opening each valve when the solenoid was energized at a 10 Hz frequency. The control system implemented in the sprayer was fed from two different power sources. On the one hand, a self-standing internal battery providing 12 V was used for computer-controlled components such as an onboard PC-104 computer, a data acquisition card (National Instruments, Austin, TX, USA), a 240 W voltage converter (12 VDC-24 VDC), a 14-inch touchscreen monitor that hosts the graphic user interface (GUI) and a GPS receiver (SXblue, Anjou, QC, Canada). On the other hand, an external source was directly connected to the pulling tractor´s battery, providing 12 V and 30 A for powering the 16 solenoids of the sprayer. A pressure sensor (Wika A10, Wika, Klingenberg, Germany) and a flow-rate sensor (Ifm Vortex SV7204, Ifm Electronic Essen, Germany) were connected to the computer. Figure 1 shows the modified blast sprayer evaluated, and Figure 2 provides a close-up of the three system manometers P1, P2 and P3 (P1 immediately after the pump, P2 before the main pipe dividing the flow between the two sides and P3 at the entrance of the right nozzle arc).

The intelligent spraying system implemented in the commercial sprayer was designed to operate under two working modes: a manual mode to analyze the specific behavior of selected groups of nozzles and denominated sectors and an automatic mode to read a prescription map and apply the prescribed rates according to the sprayer position. Sectors 1 (right) and 2 (left) comprise the four nozzles of the central part of the arc, whereas sectors 3 (right) and 4 (left) include the two top and two bottom nozzles targeting the periphery of the canopies. Figure 3 depicts a schematic of the operational modes embedded in the modified sprayer. For the evaluation reported in this paper, only the manual mode was necessary.

### 2.2. Methods

The system measured the flow rate (L/min) in the main pipe before it was divided between the two sides (left and right), together with the pressure (kPa) at three specific points in the circuit (P1, P2 and P3). As the sprayer was symmetrically configured (Figure 1), only the right side was studied in detail, considering the eight PWM nozzles distributed in sector 1 (intermediate) and sector 3 (superior-inferior). Four different duty cycles (25 percent, 50 percent, 75 percent and 100 percent) and four different pressure settings regulated at P1 (400 kPa, 500 kPa, 600 kPa and 700 kPa) were tested. The flow rate for each specific nozzle was experimentally determined by measuring the water volume collected in a container (Figure 1) during an established time interval between 180 s and 480 s. In addition, the opening and closing maneuvers of the nozzles were monitored with a high-speed camera providing a 300 fps video recording. Several experiments were run to relate the flow rate of the different nozzles to the duty cycles and pressure settings.

## 3. Results

### 3.1. Effect of Duty Cycle and System Pressure on Flow Rate

According to the ANOVA analysis provided in Table 1, both the duty cycle and the system pressure had a significant effect on the average flow rate measured for the intermediate nozzles (3 to 6) of sector 1. This sector was chosen as a representative sector for blast sprayers because the two top nozzles (sector 3) were hydraulically connected to the nozzle arc using a customized solution for this particular sprayer, and therefore sector 3 offered a less generic solution for analysis. The measured rates increased when the pressure increased and also when the percentage duty cycle increased from 25 percent to 100 percent, with higher increments found as the duty cycle increased. For a system pressure of either 500 kPa or 600 kPa, the resulting curves presented an unusual pattern that did not follow the theoretical relation between flow rate and pressure given in Equation (Equation 1) below, where *q* is the flow rate, *k* is the nozzle discharge coefficient and *p* is the pressure.
(1)$$q=k\times p^{0.5}$$

Figure 4 shows a comparison between the theoretical and the experimental flow rates for the studied nozzles according to the duty cycle (DC), for system pressures of 400 kPa and 700 kPa. The differences in the magnitudes of the flow rates may originate in the practical difficulty of reading the system pressure accurately near the nozzles at high pulsing regimes, in addition to the load losses related to the equipment design. Higher discrepancies were found at 25 percent DC for 400 kPa, and 50 percent and 75 percent DC for 700 kPa. Theoretical values were always above real measurements, since the manufacturer’s tables referred to nozzle pressure whereas these experiments registered system pressure at P1, with offsets reaching 100 kPa or above. The loss of linearity in the experimental curves was probably caused by the lack of stability in the laboratory measurements when the nozzles were pulsing below 75 percent DC. Better measuring assemblies are being designed for future tests to be performed both in the laboratory and in olive groves.

According to Figure 5, the reduction in flow rate when DC decreases, which is the goal of PWM nozzle control, was different depending on the DC. In particular, for 25 percent and 50 percent DCs, the reduction did not depend on the pressure, with an approximately 19 percent reduction in flow rate obtained for 25 percent DC and a 32 percent reduction for 50 percent DC, for the range of pressures tested (400–700 kPa). However, for 75 percent DC, the reduction in flow rate at 400 kPa was considerably lower, whereas the reduction at 500 kPa was higher, giving an average flow rate of 69 percent compared to the expected 75 percent DC. Considering prior results for similar pressure ranges [24] showing that the smaller the nozzles the smaller the changes in pressure and flow rate produced, the pressure fluctuations observed in Figure 4 and Figure 5 could be due to the large size of the nozzles implemented in the sprayer, which supplied the highest flows in the manufacturer’s series, as required for crop protection in adult olive trees.

### 3.2. System Pressure Recorded by the Control System

The instantaneous pressure registered by the pressure sensor connected to the computer was compared to the pressure gauge value (Figure 2) read after the pump at P1 by visual inspection, and plotted in Figure 6. As expected, a good linear relation was found (R2 = 82 percent), although some discrepancies appeared between 490 kPa and 590 kPa as registered by the pressure sensor, and between 500 kPa and 600 kPa as measured by the gauge. The average difference between the pressures measured by the two systems was only 18 kPa, with a maximum divergence of 94 kPa, which could easily be due to a visual error when reading the pressure gauge.

A simple regression model was built to correlate the sector 1 flow rate with the pressure estimates obtained from the two registering systems, i.e., the computer-connected electronic sensor and the pressure gauge. The best model and the linear model fitting are shown in Table 2.

Based on Table 2, the best R2 values were found for the pressure registered by the computer for all the duty cycles tested. Specifically, the R2 values were considerably higher for 100 percent and 75 percent DC (R2 > 70 percent) but noticeably lower for 50 percent and 25 percent DC (R2 < 60 percent for 50 percent DC and R2 < 10 percent for 25 percent DC). However, when considering all the duty cycles together, no clear relation between pressure and flow rate was found. In consequence, a deeper analysis was required, and thus a multiple regression model was built to relate flow rate, Psensor and DC, as depicted in Figure 7. The results for this model, given in Table 3, show that it significantly outperforms the previous one, with an R2 of 93.1 percent, where both independent parameters meaningfully affect the measured flow rate (*p*-value < 0.05). The regression equation is given in Equation (Equation 2).
(2)Flow rate(L/min)=[−1.7880−0.2072∗Psensor+0.0490∗DC]

### 3.3. Flow Rate Recorded by the Sprayer Computer with an Electronic Flowmeter

The instantaneous flow rate registered by an electronic flowmeter connected to the control computer was compared to the manually measured flow rates (Figure 1). A weak correlation was found with a linear model R2 of 47 percent and a best model R2 of 53 percent. After removing some discrepant data from the model, in particular data from test number 7 and nozzles 1 and 2 (sector 3 with unconventional pipe arrangement), the correlation improved considerably, up to a linear model R2 of 73 percent and a best model R2 of 85 percent. However, the overall relation between the manually measured flow rate and the computer-registered flow rate was not accurate enough. In order to assess the effect of the actual opening period of each DC on the flow measurement discrepancies, the correlation between both flow rates was studied for every duty cycle after removing the data from test 7 and nozzles 1 and 2. The output given in Table 4 indicates that it is necessary to consider the amplitude of the pulse cycle to obtain a good relation between the actual flow rate and the computer flow rate. As the flow fluctuations produced by the pulses near the nozzle arc affect the measuring capacity of the flowmeter, it is crucial to improve the current design by reducing these perturbances in the sensing sections of the spraying circuit before registering accurate flow-rate data in real time.

### 3.4. Evaluation of PWM Nozzles

Based on the high-speed camera images of the nozzles at 50 percent DC, the durations of the duty cycles were accurately calculated by registering the initial and final time frames for the footage sections analyzed. Table 5 provides the numerical results of the time-lapse calculations, and Figure 8 shows three snapshots of one nozzle at its opening (a), duty (fully open) (b) and closing (c) times. These results confirmed that the pulse frequency was 10 Hz (as commanded by the control computer) and the pulse duration was 55 percent on average for a set duty cycle of 50 percent. This insignificant difference was probably the result of an imprecise selection of the final frame in the footage.

In order to assess and understand the pressure variations at the nozzle location, the pressure measurements obtained by the three conventional manometers in Figure 2 were compared when the pre-set system pressure at P1 was 600 kPa, as detailed in Table 6. When the DC was 100 percent, and therefore the nozzles remained continually open, the pressure variations in each gauge were low (less than 40 kPa), with a pressure drop of 8 kPa between the P1 and P2 positions and almost 200 kPa between the P1 and P3 positions, i.e., from the pump outlet to the entrance of the nozzle right arc. For a DC of 75 percent and below, fluctuations increased as the DC decreased, and the pressure gauge location approximated the nozzles, with the most extreme case being the manometer at P3 for a DC of 50 percent, with peak pressures of 0 kPa and 1140 kPa yielding an oscillatory range of 1140 kPa when the pre-set pressure at P1 was 600 kPa. Such intense pressure oscillations at nozzle level suggest the need to devise more sophisticated methods to determine droplet formation as a function of system pressure build-up under a pulsing scenario at 10 Hz, to regulate the flow rate.

## 4. Conclusions

Pulse-width-modulated solenoid-driven nozzles mounted in a commercial blast sprayer were capable of modulating the flow rate according to the duty cycle under static conditions, although several complexities arose in the laboratory experiments. As expected, the flow rates increased with system pressure and the extent of the DC from 25 percent to 100 percent, but this increment was lower than expected from theoretical estimations. Specifically, for 25 percent and 50 percent DC, the actual reduction in flow was 19 percent and 32 percent, respectively. In terms of monitoring the relevant parameters of the spraying system, the instantaneous measurement of pressure with a sensor connected to the main computer provided accurate registers, but the real-time measurements of flow rates with an in-line flowmeter gave unreliable results due to the severe oscillations in the flowmeter line induced by the pulsing nozzles. A better location and attenuation of the flow line measuring point would probably eliminate, or significantly reduce, the errors found in the real-time estimation of flow rates. In contrast, the nozzle actuation timing tracked by high-speed video recording was very precise, confirming the selected 10 Hz frequency and yielding time lapses very close to the required DC. An important pressure drop of 200 kPa was measured between the outlet of the pump (P1) and the beginning of the right nozzle arc (P3), which indicates the convenience of always distinguishing between system pressure and nozzle pressure for follow-on studies. This study showed the inaccuracies in flow-rate reduction according to the fixed duty cycle and the difficulties of measuring nozzle pressures at low DC, with the resulting consequences for stable flow rate regulation and droplet-size homogeneity. Overall, PWM-actuated nozzles offer an attractive pressure-flow control option for smart spraying, but there are still important technical challenges that need thorough analysis before the widespread adoption of this technology for orchards and specialty crops.

## Figures and Tables

**Figure 1 sensors-22-04924-f001:**
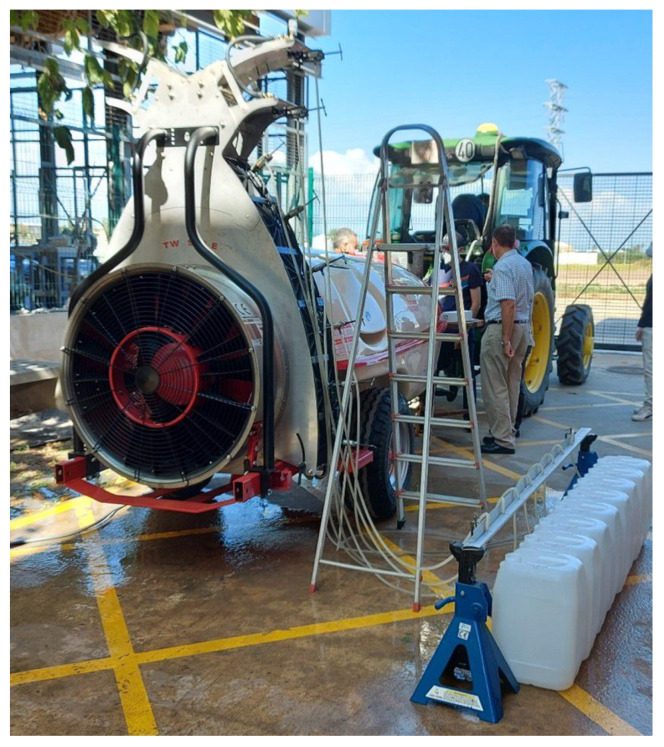
Laboratory evaluation of a modified blast sprayer.

**Figure 2 sensors-22-04924-f002:**
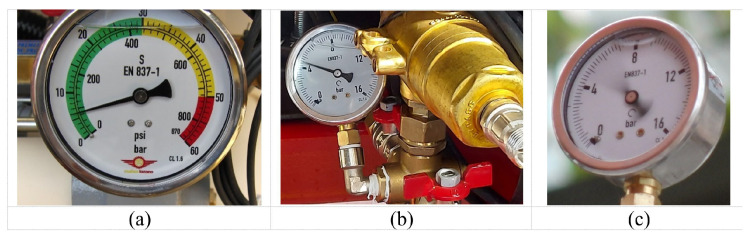
Pressure measurements at key circuit points P1 (a), P2 (b) and P3 (c).

**Figure 3 sensors-22-04924-f003:**
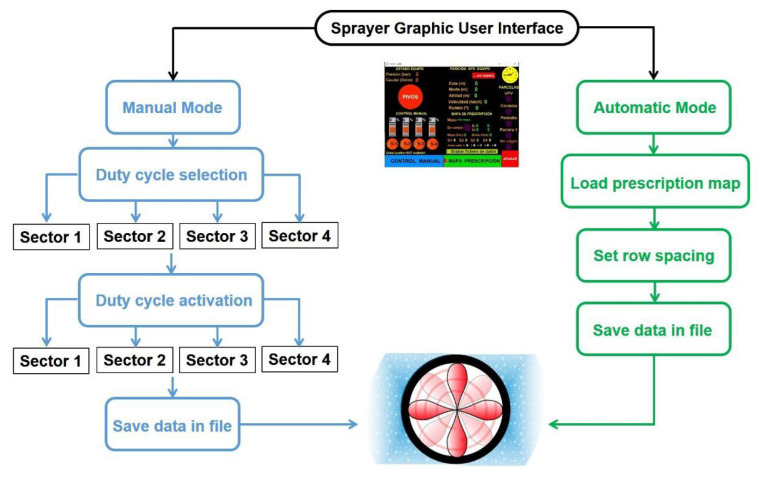
Working modes in smart sprayer: manual mode (left) and automatic mode (right).

**Figure 4 sensors-22-04924-f004:**
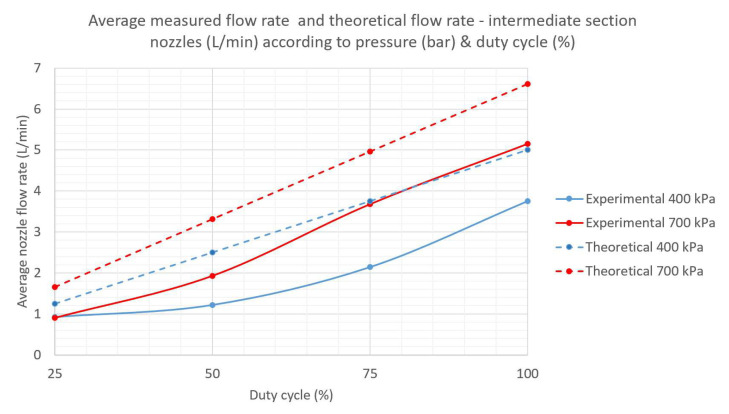
Experimental and theoretical flow rates in sector 1 nozzles (L/min), according to system pressure at P1 (kPa) and DC (percent).

**Figure 5 sensors-22-04924-f005:**
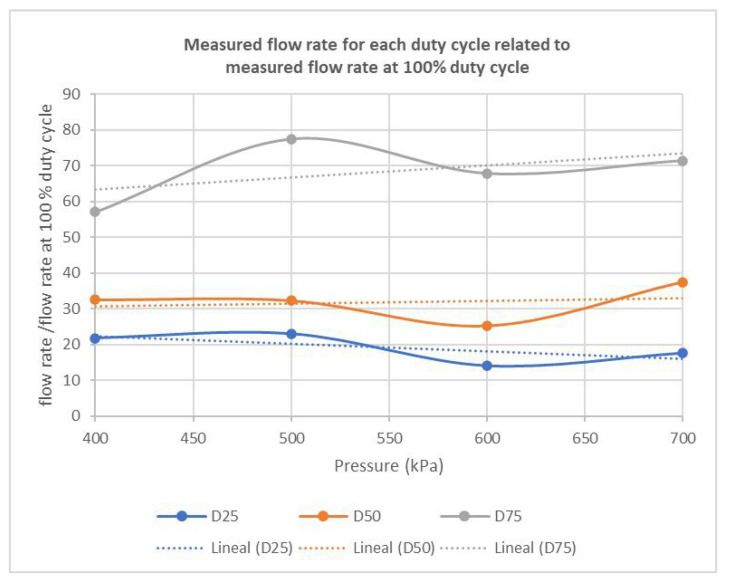
Measured flow rate related to the flow rate at 100 percent DC, for pressures and DCs.

**Figure 6 sensors-22-04924-f006:**
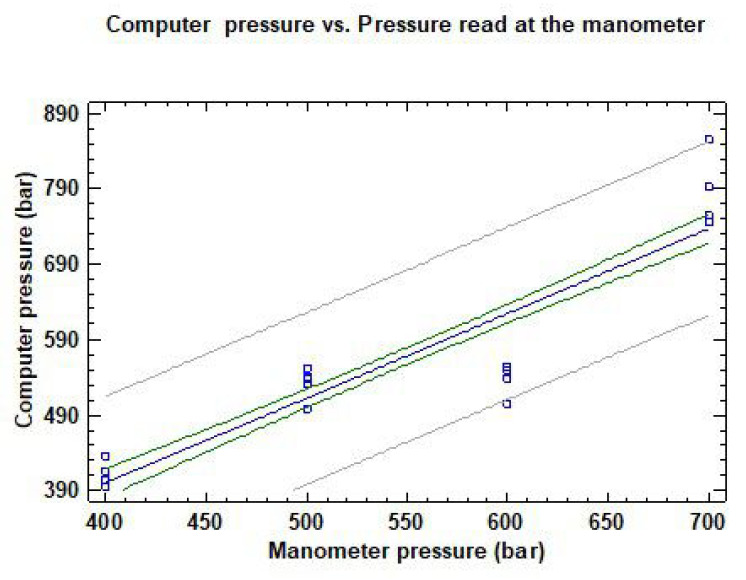
Pressure registered by the computer compared to the pressure read at the manometer gauge at P1.

**Figure 7 sensors-22-04924-f007:**
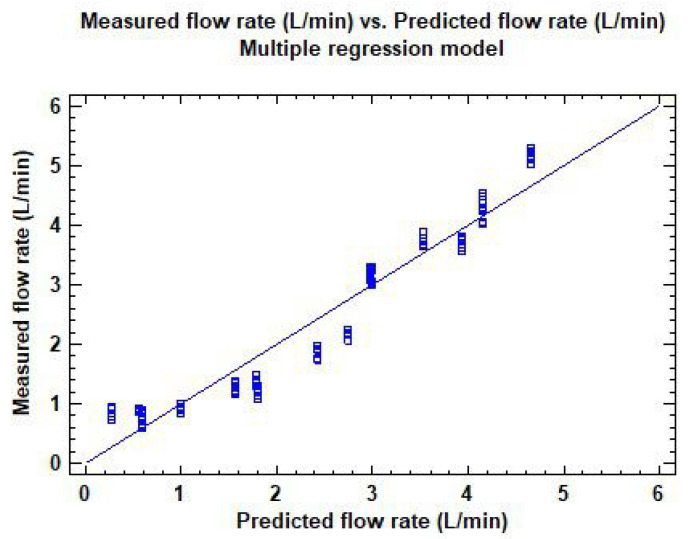
Predicted flow rate vs. measured flow rate for the multiple regression model in Equation (Equation 2) and Table 3.

**Figure 8 sensors-22-04924-f008:**
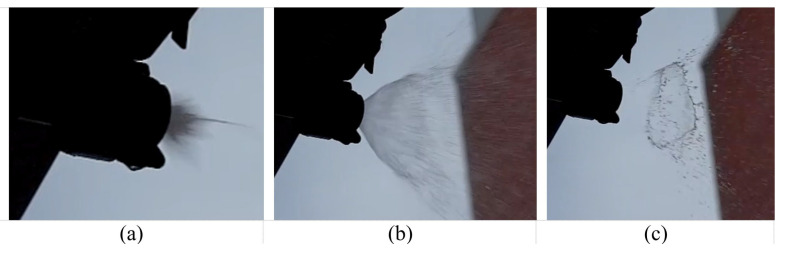
Duty cycle pulses: (**a**) opening point, (**b**) duty (fully opened) and (**c**) closing point.

**Table 1 sensors-22-04924-t001:** Two-way analysis of variance for duty cycle, system pressure, and their interaction in the measured flow rates of sector 1 nozzles.

Source	Sum of Squares	df	Mean Square	F-Ratio	*p*-Value
Duty cycle	35.85	3	11.95	418.31	0.0000
System pressure	1.94	3	0.65	22.60	0.0057
Duty cycle × System pressure	1.17	9	0.13	4.55	0.0794
Residuals	0.11	4	0.03		
Total	52.45	19			

**Table 2 sensors-22-04924-t002:** R2 (percent) from the simple regression models (best model and linear model) between pressure (pressure registered by the computer (Psensor) and pressure measured by the reference gauge after the pump (P1)) and flow rate.

Duty Cycle	Q vs. P1	Q vs. Psensor	Q = k.P1	Q = k.Psensor
	R2 Best Model	R2 Best Model	R2 Linear Model	R2 Linear Model
All duty cycles	8	2	5	2
100	91	91	89	90
75	84	91	78	74
50	34	59	21	43
25	1	9	0	6

**Table 3 sensors-22-04924-t003:** Multiple regression model for the flow rate as a function of Psensor and DC.

R2 = 93.10 Percent		Error	Statistic	
Parameter	Estimation	Standard	T	*p*-value
CONSTANT	−1.7880	0.1874	−9.5394	0.0000
PSensor (kPa)	0.2072	0.0282	7.3586	0.0000
Duty cycle	0.0490	0.0013	36.6017	0.0000

**Table 4 sensors-22-04924-t004:** Linear relation between measured flow rate (L/min) and computer-registered flow rate (Sensor flow rate) for every duty cycle.

Duty Cycle	R2	Model
100	97	Flow rate (L/min) = −0.1712 + 1.1111 × Sensor flow rate
75	84	Flow rate (L/min) = −0.6370 + 1.1105 × Sensor flow rate
50	46	Flow rate (L/min) = 0.3992 + 0.3625 × Sensor flow rate
50´	87	Flow rate (L/min) = 0.3452 + 0.4186 × Sensor flow rate
25	65	Flow rate (L/min) = 1.0436 − 0.1424 × Sensor flow rate

**Table 5 sensors-22-04924-t005:** Calculation of duty cycle duration from camera images at 50 percent duty cycle.

Frame 0	Frame f	Duration (s)	Period (s)	Duty Cycle
127	144	0.057	-	57
157	173	0.053	0.1	53
127	187	0.057	0.1	57
127	217	0.057	0.1	57
127	247	0.053	0.1	53
127	277	0.057	0.1	57
Mean	-	-	0.1	55

**Table 6 sensors-22-04924-t006:** Pressure measurements read at P1, P2, and P3 gauge positions.

DC		P1			P2			P3		
Max	Min	Range	Max Min	Range	Max Min	Range				
100	6.0	-	-	5.4	5.0	173	0.4	4.2	4.0	0.2
75	6.2	6.0	0.2	5.6	4.2	173	1.4	6.5	2.1	4.4
50	6.0	5.8	0.2	8.6	3.6	173	5.0	11.4	0.0	11.4
25	6.0	-	-	7.8	3.6	173	4.2	10.5	0.5	10.0

## Data Availability

Not applicable.

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
