# Peer review of "Preliminary Evaluation of a Blast Sprayer Controlled by Pulse-Width-Modulated Nozzles"

_sensors, 2022, doi:10.3390/s22134924_

Round 1
Reviewer 1 Report
The manuscript discusses the practical feasibility and preliminary evaluation of a blast sprayer controlled by pulse width modulated nozzles. Pulse-width-modulated solenoid-driven nozzles mounted in a commercial blast sprayer can be used to modulate flow rate in according to duty cycle under static condition. The study was well carried out with background information, review of used cases in area of pulse width modulation for spray application. The discussions shows clearly the contribution of the feasibility study. However, here are my comments and suggestions towards improving the content of the manuscript
Line 62: I suggest that “Actuation classes, for…” should be “actuation classes. For…”
82: flowrates – flow rate, 85 flow rate (for uniformity due to constant use of “flow rate” in the paper)
103: I suggest that “to use” should be changed to “using or the usage of” for easier reading
Section 2 on Materials and methods… I would suggest that the materials are listed separately before going into the methods in which the materials are used for the sake of being distinct and for a clearer understanding and implementation by future readers.
In figures 4 and 7 (title of the plot): the word “ meassured” should be measured
230: sprayer, may be changed to sprayer without comma
268: in the series a b c… b is named as duty whereas in figure 8, b is named as “fully opened”… check
274: table ??
308: “being thoroughly analyzed” can be changed to “ thorough analysis”
I recommend that you improve on the quality of all your figures. Make it more clearer
The tables and the data in them should be checked as well i.e Table 5, 6, some of the content are missing. I suggest you use – (dash) in the cells where there is no value.
Author Response
Responses to Reviewer 1
The manuscript discusses the practical feasibility and preliminary evaluation of a blast sprayer controlled by pulse width modulated nozzles. Pulse-width-modulated solenoid-driven nozzles mounted in a commercial blast sprayer can be used to modulate flow rate in according to duty cycle under static condition. The study was well carried out with background information, review of used cases in area of pulse width modulation for spray application. The discussions shows clearly the contribution of the feasibility study. However, here are my comments and suggestions towards improving the content of the manuscript
- Thank you very much for your clear comments on the manuscript. All your suggestions have been carefully addressed.
Line 62: I suggest that “Actuation classes, for…” should be “actuation classes. For…”
- It was modified according to the reviewer suggestion, but this part has been removed according to another reviewer comment.
82: flowrates – flow rate, 85 flow rate (for uniformity due to constant use of “flow rate” in the paper)
- “Flowrate” and “flowrates” have been modified in the paper to “flowrate” and “flow rates” in all the cases.
103: I suggest that “to use” should be changed to “using or the usage of” for easier reading
- According to the reviewer, “to use” has been changed by “using” (new line 80).
Section 2 on Materials and methods… I would suggest that the materials are listed separately before going into the methods in which the materials are used for the sake of being distinct and for a clearer understanding and implementation by future readers.
- The materials have been listed separately before the methods. Two subsections have been included “Materials” and “Methods”
In figures 4 and 7 (title of the plot): the word “ meassured” should be measured
- The mistake has been corrected
230: sprayer, may be changed to sprayer without comma
- Corrected (new line 210).
268: in the series a b c… b is named as duty whereas in figure 8, b is named as “fully opened”… check
- It has been checked and corrected, in the text and in the Figure 8 it has been written “duty (fully opened)”, new lines 247-248 and figure 8.
274: table ??
- Corrected (new line 254).
308: “being thoroughly analyzed” can be changed to “ thorough analysis”
- Corrected (new line 289).
I recommend that you improve on the quality of all your figures. Make it more clearer
- The quality of the figures has been improved.
The tables and the data in them should be checked as well i.e Table 5, 6, some of the content are missing. I suggest you use – (dash) in the cells where there is no value.
- The tables have been corrected.

Reviewer 2 Report
I review the article title “Preliminary evaluation of a blast sprayer controlled by pulse width modulated (PWM) nozzles” submitted for possible publication in the Sensors. In this article, the authors investigated the performance of a commercial blast sprayer modified with pulse width modulated nozzles under laboratory conditions by considering four different duty cycles and four different pressures as a preliminary step before its further field validation. After carefully, reviewing the paper, I came to conclude that the article is not recommended for publication in Sensors. The manuscript is not presented in a well-structured manner and is clearly written. It isn’t scientifically sound and the experimental design is not appropriate to test the hypothesis. I suggest that the authors should significantly revise the introduction and material methods section; such as in the introduction, from line 51 “Different sensors have been developed and tested to obtain information about the canopy” to line 75 “time to apply the precise dose required at each location” is not necessary to mention. In this article, these methods are considered as data collection sources. Authors should focus on their main influencing parameters such as duty cycles and pressures.
Author Response
Responses to Reviewer 2
I review the article title “Preliminary evaluation of a blast sprayer controlled by pulse width modulated (PWM) nozzles” submitted for possible publication in the Sensors. In this article, the authors investigated the performance of a commercial blast sprayer modified with pulse width modulated nozzles under laboratory conditions by considering four different duty cycles and four different pressures as a preliminary step before its further field validation. After carefully, reviewing the paper, I came to conclude that the article is not recommended for publication in Sensors. The manuscript is not presented in a well-structured manner and is clearly written. It isn’t scientifically sound and the experimental design is not appropriate to test the hypothesis. I suggest that the authors should significantly revise the introduction and material methods section; such as in the introduction, from line 51 “Different sensors have been developed and tested to obtain information about the canopy” to line 75 “time to apply the precise dose required at each location” is not necessary to mention. In this article, these methods are considered as data collection sources. Authors should focus on their main influencing parameters such as duty cycles and pressures.
- The introduction has been modified according to the reviewer suggestion. The introduction part from “Different sensors have been developed and tested to obtain information about the canopy” to “time to apply the precise dose required at each location” has been removed.
- The objective of the study has been rewritten in order to clear the idea of studying the main influencing parameters such as duty cycles and pressures.
“The objective of this paper is to characterize the performance of a commercial blast sprayer modified with pulse width modulated nozzles under laboratory conditions, testing the effect of duty cycle and pressure setting on flow rate and pressure at different circuit point. This experiment was a preliminary step before the application of prescription maps featuring VRT actuation in vineyards and olive groves for a sustainable spraying and an efficient crop protection.”
- The conclusions have been modified to clarify the main results of the study.
“This study shows the inaccuracies in flow rate reduction according to the fixed duty cycle and the difficulties of measuring nozzle pressures at low DC, and its resulting consequences for stable flow rate regulation and droplet size homogeneity. Overall, PWM-actuated nozzles offer an attractive pressure-flow control for smart spraying, but there are still important technical challenges that need thorough analysis before the widespread adoption of this technology for orchards and specialty crops.”
Reviewer 3 Report
This manuscript reports the performance and characteristics of PWM nozzles for agricultural blast spray equipment. Although it is a suitable content category for the Advances in Control and Automation in Smart Agriculture issue of Sensors journal, many improvements are required in this manuscript.
As major comments,
First, there are many shortcomings in the originality of the study. In interpreting the characteristics of commercial blast spray, theoretical design or scientifically meaningful discovery is required. Overall, the current version of the manuscript is in the form of a report. A scientific approach is most needed.
Second, the meaning of the results of the study is lacking. It is a general result from the reviewer's point of view that the nozzle's flow rate is correlated with the increase in the duty cycle of PWM.
As minor comments,
1) The introduction is well written in detail, but it is not divided into paragraphs, and the difference between the previous study and this study is not well emphasized.
2) page 5, line 187, why is the GPS receiver mentioned?
3) All result graphs have different formats. (Figures 4,5,6,7) In particular, the pressure unit shown in Figure 6 is 'Bar', and other than that, 'kPa' is used in the text.
4) In the performance evaluation of agricultural sprayer nozzles, the results of particle size and cover area when sprayed are required for evaluation criteria.
Author Response
Responses to Reviewer 3
This manuscript reports the performance and characteristics of PWM nozzles for agricultural blast spray equipment. Although it is a suitable content category for the Advances in Control and Automation in Smart Agriculture issue of Sensors journal, many improvements are required in this manuscript.
As major comments,
First, there are many shortcomings in the originality of the study. In interpreting the characteristics of commercial blast spray, theoretical design or scientifically meaningful discovery is required. Overall, the current version of the manuscript is in the form of a report. A scientific approach is most needed.
Second, the meaning of the results of the study is lacking. It is a general result from the reviewer's point of view that the nozzle's flow rate is correlated with the increase in the duty cycle of PWM.
- Thank you very much for your comments.
- The objective of the study has been rewritten in order to clear the aim of the study, the idea of studying the main influencing parameters such as duty cycles and pressures has been remarked.
“The objective of this paper is to characterize the performance of a commercial blast sprayer modified with pulse width modulated nozzles under laboratory conditions, testing the effect of duty cycle and pressure setting on flow rate and pressure at different circuit point. This experiment was a preliminary step before the application of prescription maps featuring VRT actuation in vineyards and olive groves for a sustainable spraying and an efficient crop protection.”
- The materials have been listed separately before the methods. Two subsections have been included “Materials” and “Methods”
- The conclusions have been modified to clarify the main results of the study.
“This study shows the inaccuracies in flow rate reduction according to the fixed duty cycle and the difficulties of measuring nozzle pressures at low DC, and its resulting consequences for stable flow rate regulation and droplet size homogeneity. Overall, PWM-actuated nozzles offer an attractive pressure-flow control for smart spraying, but there are still important technical challenges that need thorough analysis before the widespread adoption of this technology for orchards and specialty crops.”
As minor comments,
1) The introduction is well written in detail, but it is not divided into paragraphs, and the difference between the previous study and this study is not well emphasized.
- The introduction has been reduced and divided in paragraphs.
- The objective of the present study has been remarked at the end of the introduction.
2) page 5, line 187, why is the GPS receiver mentioned?
- The reference to the GPS has been removed.
3) All result graphs have different formats. (Figures 4,5,6,7) In particular, the pressure unit shown in Figure 6 is 'Bar', and other than that, 'kPa' is used in the text.
- Figure 6 has been corrected.
4) In the performance evaluation of agricultural sprayer nozzles, the results of particle size and cover area when sprayed are required for evaluation criteria.
- We completely agree with the reviewer. Further research should be carried out to study particle size related to duty cycle. Future experiments are being designed in this line.
Round 2
Reviewer 2 Report
I am sorry for giving you short comments that this manuscript (MS) is not recommended for possible publication in the Sensors journal.
Author Response
The first version of the manuscript was presented at the III Iberian Symposium of Horticultural Engineering 2022, Smart Farming, and received the Symposium Prize Best Paper Award.
We consider that the new revised version of the manuscript could be publish in the Sensors special issue of the conference "Advances in Control Automation in Smart Agriculture".
Reviewer 3 Report
The authors have already addressed all reviewer comments properly. In the academic field, it is expected that there will be many improvements in future research.
- Please increase the font of the text in Figure 4.
Author Response
Thank you for your comments.
The font in Figure 4 has been increased (we attach new Figure4Corrected2.jpg).
